# Influence of observer-dependency on left ventricular hypertrabeculation mass measurement and its relationship with left ventricular volume and ejection fraction – comparison between manual and semiautomatic CMR image analysis methods

**Marcin Kubik**[1☉], **Alicja Dąbrowska-Kugacka**[1☉]*, **Karolina Dorniak**[2‡], **Marta Kutniewska-Kubik**[3‡], **Ludmiła Daniłowicz-Szymanowicz**[1], **Ewa Lewicka**[1], **Edyta Szurowska**[4], **Grzegorz Raczak**[1]

1 Department of Cardiology and Electrotherapy, Medical University of Gdansk, Gdansk, Poland,
2 Department of Cardiac Diagnostics, Medical University of Gdansk, Gdansk, Poland, 3 Centre of Psychological Diagnosis, Therapy, and Personal Development, Mala Piasnica, Poland, 4 Department of Radiology, Medical University of Gdansk, Gdansk, Poland

☉ These authors contributed equally to this work.
‡ These authors also contributed equally to this work.
* alidab@gumed.edu.pl

## Abstract

### Background

Recent studies concerning left ventricular noncompaction (LVNC) suggest that the extent of left ventricular (LV) hypertrabeculation has no impact on prognosis. The variety of methods of LV noncompacted myocardial mass (NCM) assessment may influence the results. Hence, we compared two methods of NCM estimation: largely observer-independent Hautvast's$_{(H)}$ computed algorithm-based approach and commonly used Jacquier's$_{(J)}$ method, and their associations with LV end-diastolic volume (EDV) and ejection fraction (EF).

### Methods

Cardiac magnetic resonance images of 77 persons (45±17yo) - 42 LVNC, 15 non-ischemic dilative cardiomyopathy, 20 control group were analyzed. LVNC patients were divided into the subgroup with normal (LVNC$_N$) and high EDV (LVNC$_{DCM}$). NCM and total left ventricular mass (LVM) were estimated by Hautvast's [excluding intertrabecular blood (ITB) and including papillary muscles (PMs) into NCM] and Jacquier's approach (including ITB and PMs, if unclearly distinguished, into NCM).

### Results

The cut-off value of NCM for LVNC diagnosis was 22% (AUC 0.933) for NCM$_H$/LVM$_H$ and 26% (AUC 0.883) for NCM$_J$/LVM$_J$. Inter- and intra-observer variability (estimated by

**Data Availability Statement:** All relevant data are within the paper and its Supporting Information files.

**Funding:** The authors received no specific funding for this work.

**Competing interests:** The authors have declared that no competing interests exist.

coefficient of variation [CoV] and intraclass correlation coefficient [ICC]) of $NCM_H/LVM_H$ appeared better than of $NCM_J/LVM_J$ (CoV 4.3%, ICC 0.981 and CoV 4.9%, ICC 0.978; respectively for $NCM_H/LVM_H$, while for $NCM_J/LVM_J$: CoV 19.7%, ICC 0.15 and CoV 12.9%, ICC 0.504). In $LVNC_N$ subgroup, the correlation between EDV and $NCM_H$ was stronger than $NCM_J$ ($r = 0.677$, $p<0.001$ vs. $r = 0.480$, $p = 0.038$; respectively). In LVNC the EDV correlated with $NCM_H/LVM_H$ ($r = 0.391$, $p<0.01$), but not with $NCM_J/LVM_J$. In the overall group a relationship was present between EF and $NCM_H/LVM_H$ ($r = -0.449$, $p<0.001$), but not $NCM_J/LVM_J$. Only $NCM_H/LVM_H$ explained the variability of EDV (b 0.434, $p<0.001$).

## Conclusions

Choosing a method of NCM assessment that is less observer-dependent might increase the reliability of results. The impact of method selection on the LV parameters and cut-off values for hypertrabeculation should be further investigated.

## Introduction

Left ventricular noncompaction (LVNC) is so far considered to be a unique inherited cardiomyopathy [1]. It is characterized by a spongy morphological appearance of a left ventricular (LV) myocardium with a mesh of prominent trabeculae separated by deep intertrabecular recesses [2,3]. The LV hypertrabeculation, however, can be present in healthy individuals, as well as in cardiomyopathies [4,5].

Clinically, LVNC is associated with an increased risk of cardiovascular events similar to non-ischemic dilative cardiomyopathy (nDCM) [1,3]. The LV end-diastolic volume (EDV) and ejection fraction (EF) may be the significant markers of adverse outcomes in LVNC [1,4]. In turn, the clinical significance of the LV hypertrabeculation is unclear, and some studies indicate that it is not a prognostic factor of adverse cardiovascular outcomes [1,4,6]. These observations may be related to the variety of different criteria for LVNC recognition using cardiac magnetic resonance (CMR) imaging [4,7]. They are mostly based on the estimation of a thickness ratio between LV noncompacted and compacted layers or of a mass ratio between an LV noncompacted myocardial mass (NCM) and total LV mass (LVM) [8,9,10]. The methods assessing the mass ratio differ in the approach of in- or exclusion of intertrabecular blood pool (ITB) and papillary muscles (PMs) from NCM. Such an approach may affect the LVNC diagnosis and the assessment of the influence of LV hypertrabeculation on EDV and EF. A brief review of some diagnostic LVNC criteria is shown in Table 1.

Jacquier's method of LVNC recognition adds ITB into NCM, which might falsely augment the real estimate of the latter. Additionally, it gives a possibility to include PMs either into the LV compacted layer mass (CLM) or into the LV trabeculation area if not clearly distinguished. Such a non-uniform approach to NCM estimation may decrease its reproducibility, as PMs in LVNC are often multiple and fragmented, and hence, their inclusion in either of the two layers can be equivocal. An algorithm differentiating ITB from NCM was proposed and described by Hautvast et al. [12] and is currently available as part of Philips' proprietary analysis software for the LV volumes and masses. This algorithm enables the exclusion of ITB from NCM, but so far, its value in LVNC diagnosis was not confirmed.

Our study aimed to compare the two different methods of measurement of NCM and its percentage of LVM: proposed by Jacquier et al. [10] and by Hautvast's [12] computed algorithm, and evaluate their possible impact on EDV and EF.

**Table 1. Review of the most popular left ventricular noncompaction (LVNC) recognition criteria using cardiac magnetic resonance (CMR) examination.**

Established by Petersen et al. [8]:
 1. NC/C ratio $\geq$ 2.3 (end-diastole, long-axis views)

Established by Grothoff et al. [9]:
 1. LV noncompacted mass $> 15 g/m^2$
 2. LV noncompacted mass $>$ 25% of the total LV mass
 3. Trabeculation in basal segments of LV and NC/C of $\geq$ 3:1
 4. Methodology:
 • primary LVNC recognition based on echocardiographic Jenni et al. criteria [11]
 • implementation of the CAAS MRV post-processing software (Pie-Medical Imaging, Maastricht, Netherlands) for contouring myocardial layers
 • exclusion of blood pool from the noncompacted mass
 • inclusion of the papillary muscles in the compacted myocardial mass
 • criteria established in LVNC patients without LGE

Established by Jacquier et al. [10]:
 1. Trabeculated LV mass $>$ 20% of LV global mass
 2. Methodology:
 • primary LVNC recognition based on echocardiographic criteria by Jenni et al. [11] and CMR criteria by Petersen et al. [8]
 • implementation of the Argus™ post-processing software (Siemens) for contouring myocardial layers
 • inclusion of the papillary muscles in the compacted myocardial mass, however, with the possibility of their inclusion into the trabeculation area if not clearly distinguished
 • inclusion of blood pool into the LV noncompacted mass
 • in case of a highly trabeculated LV, the assessment of global LV mass was performed by positioning the endocardial contour at the outer edges of the trabeculation net

BSA–body surface area; LV–left ventricle, NC/C–noncompacted/compacted ratio; LGE–late gadolinium enhancement

## Materials and methods

### Study design

The study was planned and performed following the European Association of Cardiovascular Imaging (EACVI) cardiac diagnostics guidelines and the Polish National Health Fund. Thus, the CMR scans were conducted as part of the standard out- and inpatients cardiac diagnostic process.

After receiving the written consent of the department heads of radiology and cardiology, respectively, the computed academic medical database records (available only in the Hospital of the Medical University of Gdansk, Poland) from 2011 to 2018 were searched for clinical data of previously examined with CMR patients with the pre-determined clinical diagnosis of LVNC, nDCM, and those without any cardiac disease (a control group). LVNC diagnosis was made based on the high clinical pre-test probability combined with structural findings assessed by two imaging methods: transthoracic echocardiography and CMR. The clinical pre-test probability was considered high in the presence of the LVNC-related symptoms (e.g., dyspnea, syncope, arrhythmia), and/or unexplained primary impaired LV function, and/or family history of cardiomyopathy. The echocardiographic LVNC criteria were adopted according to Jenni et al.: (i) no coexisting cardiac abnormalities, (ii) a two-layer structure of the LV muscle with a mesh of prominent trabeculae separated by deep perfused intertrabecular recesses (color Doppler) and the ratio of the thick noncompacted endocardium to the thinner compacted epicardium $>$ 2, (iii) the predominant localization of pathology located distal to PMs–the apex, lateral and/or inferior [11]. The CMR diagnosis of LVNC was made based on Petersen's criterion of noncompacted to compacted layer thickness ratio $>$2.3 in long axes views [8]. Patients over age 35 had either additional non-invasive or invasive investigation of

coronary artery disease. The nDCM diagnosis was made based on the global LV function impairment (EF <40%) and LV dilatation (EDV >117% of the normal values for age and sex) [13]. No genetic tests were performed. The control group consisted of patients with EF and EDV in the normal range and without any radiological or clinical evidence of cardiac disease. To compare LVNC patients with enlarged LV (LVNC$_{DCM}$) with nDCM, the LVNC group was divided into subgroups: LVNC with normal range LV (LVNC$_N$) and LVNC$_{DCM}$. The group inclusion criteria are listed in Table 2.

## Acquisition and analysis of CMR data

All patients underwent CMR using a 3.0 T scanner (Philips Achieva, Philips BV Eindhoven, The Netherlands) with a 32-channel phased-array receiver coil with repeated breath-holds. The segmented steady-state free-precession sequence was used to acquire cine images of the LV in two-, three-, and four-chamber views as well as in short-axis views to obtain a stack of contiguous short-axis slices to include the entire LV with a slice thickness of 8 mm and 2 mm gaps, according to standardized protocols [14]. The parallel acquisition technique, with an acceleration factor of 2, was used. The short-axis cine stack was analyzed semi-automatically with the use of the Philips Extended MR Workspace cardiac software package.

Epi- and endocardial contours were placed for each slice from the level of the mitral valve down to the apex. If necessary, the endocardial and epicardial contours were manually corrected.

The amount of NCM and LVM was estimated by two methods, according to Jacquier ($_J$) and utilizing Hautvast's ($_H$) computed algorithm incorporated in the Philips' proprietary software for LV masses analysis [10,12]. NCM and LVM estimated by Jacquier's method were called NCM$_J$, LVM$_J$, and NCM$_J$/LVM$_J$. [10] Adequately, NCM$_H$, LVM$_H$, and NCM$_H$/LVM$_H$ were calculated using Hautvast's algorithm. [12]

**Table 2. Left ventricular noncompaction (LVNC), nonischemic dilative cardiomyopathy (nDCM) and control group inclusion criteria.**

1) LVNC group:
  a. Petersen at al. CMR criterion of NC/C >2.3 in long axes views [8]
  b. CMR confirmation criterion of NCM$_J$/LVM$_J$ with a modified cut-off value of >31%
  c. No coronary artery disease

2) LVNC subgroup with normal LV (LVNC$_N$):
  a. EDV < 117% of URL by age and sex, by Kawel-Boehm et al. [13]
  b. EF > 40%
  c. Fulfilled LVNC group criteria

3) LVNC subgroup with enlarged LV (LVNC$_{DCM}$):
  a. EDV > 117% of URL by age and sex, by Kawel-Boehm et al. [13]
  b. EF ≤ 40%
  c. Fulfilled LVNC group criteria

4) nDCM group:
  a. EDV > 117% of URL by age and sex, by Kawel-Boehm et al. [13]
  b. EF < 40%
  c. Unfulfilled LVNC group criteria
  d. No coronary artery disease

5) Control group:
  a. EDV < 100% of URL by age and sex, by Kawel-Boehm et al. [13]
  b. EF > LRL, by Kawel-Boehm et al. [13]
  c. Unfulfilled LVNC group criteria
  d. Cardiac disease excluded

CMR–cardiac magnetic resonance; LV–left ventricle, EDV–LV end-diastolic volume; EF–LV ejection fraction; NCM$_J$/LVM$_J$—noncompacted/compacted LV layer mass ratio m. Jacquier et al. [10]; URL–upper range limit; LRL–lower range limit

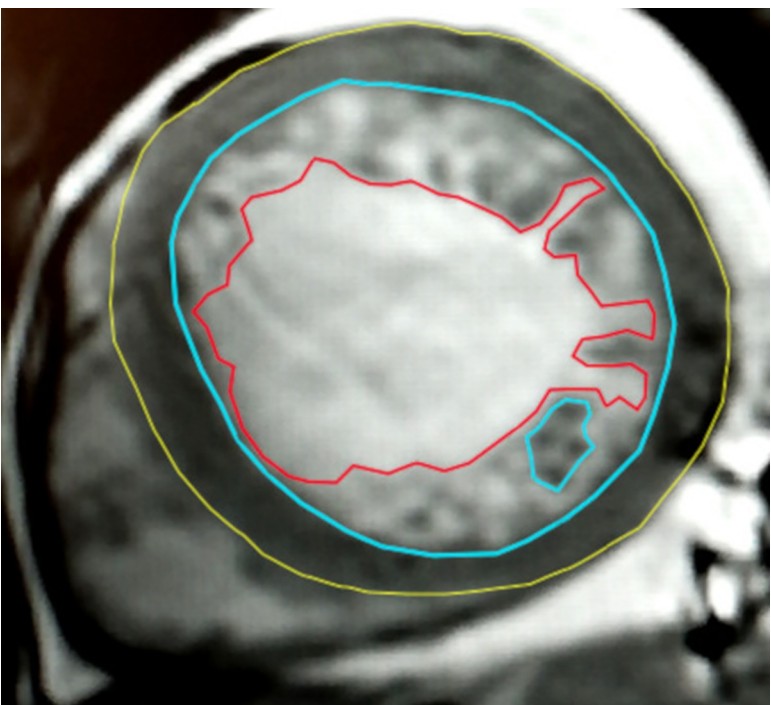

**Fig 1. Diagnostic scheme of a noncompacted layer mass assessment by Jaquier's method.** Contours: epicardial (yellow), endocardial (blue), inner endocardial (red); the left ventricular compacted layer is between the epicardial (yellow) and endocardial (blue) contours, and the noncompacted layer is between the endocardial (blue) and inner endocardial (red) contours.

In brief, differences between methods concerned the observer-dependent inclusion of the PMs' mass in CLM and the observer-independent inclusion of the ITB's mass in NCM.

**The estimate of $CLM_J$, $NCM_J$, and $NCM_J/LVM_J$.** According to Jacquier's approach, three contours were traced in all slices of the LV short-axis view in end-diastole (Fig 1):

i.  an epicardial–it delineated the outer edge of the LV compacted layer, and also delimited the volume combined with EDV and the volume of the LV compacted layer,

ii. an endocardial–it delineated the inner edge of the LV compacted layer and also delimited the standard EDV (depending on the PMs' fragmentation),
    Thus, $CLM_J$ was calculated as the difference between these two upper mentioned volumes multiplied by the density of the heart muscle (γ 1,05g/dl). In this method, PMs' mass was OPTIONALLY included in $CLM_J$ unless they were excessively fragmented, depending on the opinion of the observer and treated as trabeculation.

iii. an inner endocardial–it delineated the inner edge of the LV trabecular layer, and thus, set an observer-dependent conventional border between the LV trabecular layer and the LV cavity without trabeculation; it also delimited only the volume of the LV blood pool but without ITB.

Thus, $NCM_J$ was calculated as the volume difference between the EDV delimited by the endocardial contour and the volume delimited by the inner endocardial contour, multiplied by the density of cardiac muscle. ITB's mass was ABSOLUTELY included in $NCM_J$. In consequence, $NCM_J$ consisted of two to three masses: (i) the trabeculae, (ii) ITB, and OPTIONALLY

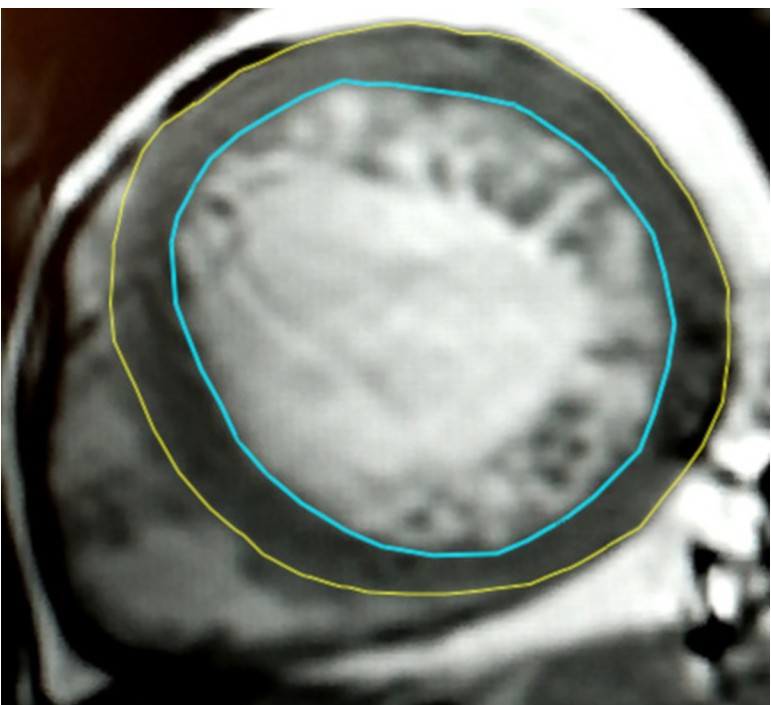

**Fig 2. Diagnostic scheme of a noncompacted layer mass (NCM) assessment by Hautvast's computed algorithm method.** Contours: epicardial (yellow), endocardial (blue); the left ventricular compacted layer is between the epicardial (yellow) and endocardial (blue) contours, and the noncompacted layer is inside the space delimited by the endocardial (blue) contour, and its mass is calculated automatically by Hautvast's computed algorithm.

and OBSERVER DEPENDENTLY (iii) PMs. $LVM_J$ was calculated, adding $NCM_J$ to $CLM_J$ as follows: $LVM_J = NCM_J + CLM_J$.

**The estimate of $CLM_H$, $NCM_H$, $NCM_H/LVM_H$.** According to Hautvast's computed algorithm, only two contours were traced in all slices of the LV short-axis view in end-diastole (Fig 2):

i. the epicardial

ii. and the endocardial, both similar to the corresponding Jacquier's contours.

In this method, the $CLM_H$ was calculated similarly to the $CLM_J$ as the volume difference between two volumes delimited by the epicardial end endocardial contours multiplied by the density of the heart muscle (γ 1,05g/dl). In opposition to Jacquier's method, however, after manual correction of the endocardial contour, PMs' mass was UNCONDITIONALLY excluded from $CLM_H$ and automatically included in $NCM_H$.

In turn, $NCM_H$ was estimated utilizing the postprocessing Philips' software as follows:

i. first, the standard EDV was obtained based on the endocardial contour location in all short-axis view slides,

ii. subsequently, $EDV_H$ was estimated using the Hautvast's algorithm [11], based on the difference in signal intensity of blood and myocardial muscle inside the endocardial contour.

As a result, ITB was ABSOLUTELY excluded from the LV trabecular layer and became part of the LV blood pool. Consistently, the $EDV_H$ represented the volume of blood pool inside the

**Table 3. The detailed comparison between the two methods of the trabecular mass measurement: By Jacquier et al. and by the semi-automatic Hautvast's algorithm implemented into Philip's CMR software [10,12].**

| Method | Hautvast's | Jacquier's |
|---|---|---|
| CLM | papillary muscles *absolutely not included* | papillary muscles *included* unless excessively fragmented* |
| LV papillary muscles | *absolutely included* in the NCM | *not included* in the NCM unless excessively fragmented* |
| LV intertrabecular blood pool mass | *absolutely excluded* from the NCM (it *was* the part of the LV blood volume) | *absolutely included* in the NCM (it *was not* the part of the LV blood volume) |
| NCM | contains:<br>• the trabecular mass<br>• the LV papillary muscles mass | contains:<br>• the trabecular mass<br>• the LV intertrabecular blood pool mass<br>• the LV papillary muscles, however, *only if* excessively fragmented* |
| estimate of NCM | Algorithm | Observer |
| number of contours | 2 | 3 |
| epicardial contour position | on the outer edge of the LV compacted layer | on the outer edge of the LV compacted layer |
| endocardial contour position | on the inner edge of the LV compacted layer, however, separating the papillary muscles from the LV compacted layer | on the inner edge of the LV compacted layer, also covering the LV papillary muscles, and thus, including them in the LV compacted layer |
| the interior endocardial contour position | not applicable | on the top of the LV trabeculae, thus, delimiting the LV noncompacted layer from the LV cavity |

LV–left ventricular, CLM–LV compacted layer mass, NCM–LV noncompacted layer mass

* observer-dependence

LV at end-diastole. Thus, the $NCM_H$ consisted of ONLY two masses: of (i) the trabeculae and UNOPTIONALLY (ii) PMs, and was calculated by the formula: $EDV—EDV_H$, multiplied by the density of cardiac muscle. $LVM_H$ was calculated by adding $NCM_H$ to $CLM_H$ as follows: $LVM_H = NCM_H + CLM_H$.

The detailed comparison between methods is shown in Table 3.

**The estimate of EF.** The standard EDV and end-systolic volume (ESV) were obtained based on the endocardial border (of the compacted myocardium) contour displacement in all short-axis view slides, and EF was calculated with the formula: (EDV-ESV)/EDV * 100%. PMs and trabeculation were included as part of the LV cavity's volume.

The institutional research ethics board (The Independent Bioethics Committee for Scientific Researches by the Medical University of Gdansk; no. of consent NKEBN/41/2012) approved the study, and each study participant provided informed written consent to CMR and enrollment of biographical data into the analysis.

## Data collection and statistical analysis

Statistical analysis was performed using the licensed Statistica 13 software package (Statsoft Poland). All continuous variables are presented as mean ± standard deviations (SDs) or median with interquartile range. Categorical variables are reported as a percentage. Statistical significance was defined as $p < 0.05$. The Shapiro-Wilk test was used to estimate the distribution. The independent t-Student test (for normally distributed continuous data) and the U-Mann-Whitney test (for not normally distributed continuous data) were used to compare between two groups. Differences between categorical variables were tested with the Chi-square test. In order to determine the cut-off value for the pathological trabecular mass in our population according to Jacquier's method, apart from the mean ± SDs of $NCM_J/LVM_J$, the upper confidence interval (+95% CI) was assessed in our control group, following Amzulescu et al. [4]. The ± 95% CI was calculated to establish the cut-off values of $NCM_H$ and $NCM_H/LVM_H$ for LVNC recognition.

ANOVA or Kruskal-Wallis test, where appropriate, were performed to investigate differences among the examined groups and afterward the comparison between the subgroups of LVNC (LVNC$_N$ and LVNC$_{DCM}$) and the nDCM or the control groups were performed with post-hoc analysis. Subsequently, Pearson's correlation analyses between NCM or NCM/LVM estimated by both methods vs. EF or EDV were performed. Finally, a multivariate stepwise regression analysis was done to create the best potential model explaining the variability of EF and EDV. The potential difference between sensitivity and specificity of cut-off values for the NCM and NCM/LVM between the LVNC and the control groups was estimated by the receiver operating characteristic curve (ROC).

### Inter- and intra-observer measurements

To calculate inter-observer variability (reproducibility), NCM and NCM/LVM were calculated by both methods (Jacquier's and Hautvast's) in 10 LVNC randomly selected patients by two independent and experienced observers blinded to each other results; to assess intra-observer variability (repeatability), the two analyses were performed by the same observer in 10 LVNC patients. Coefficients of variation (CoV; as the SD of the differences divided by the mean) and intraclass correlation coefficients (ICC) were calculated. ICC was calculated using a model of the absolute agreement for intra-observer variability and consistency for inter-observer variability. The ICC's values less than 0.5 assumed indicative of poor, 0.5 to 0.75 of moderate, 0.75 to 0.9 of good, and greater than 0.9 of excellent reliability. CoV was calculated using a within-subject SD method. We assumed CoV less than 5% as excellent, 5 to 10% as good, 10 to 20% as acceptable, and over 20% as poor data compliance [15].

## Results

One hundred two examinations were extracted from the CMR database (LVNC, nDCM, the control group– 50, 21, 31, respectively), however, 23 examinations were excluded from further analysis due to death before the study qualification, doubtful diagnosis, CMR artifacts and additional cardiac diseases which could influence the group qualification. The subsequent analysis of NCM$_J$/LVM$_J$ in the control group determined the cut-off value >31% for the recognition of the pathological trabecular mass percentage (mean 24%; ±95% CI 18–31%). Thus, two of the 44 examinations from the whole LVNC group did not meet the confirmation criterion. Among 77 examinations, which were further analyzed there were 42 (54.5%) with determined diagnosis of LVNC (age 45±17y, men 47.6%), 15 (19.5%) with nDCM (age 45±19y, men 69.2%), and 20 subjects from the control group (age 48±19y, men 54.4%) %)–see S1 Fig. Clinically, in the LVNC group, 2 individuals had prior myocardial inflammation, 2 had previous transient ischemic attack, 1 had paroxysmal atrial fibrillation, 3 suffered from the 1st-grade well-controlled hypertension, 2 were diabetics, 1 was subjected to an ablation procedure due to the Wolf-Parkinson-White syndrome, 1 had the atrioventricular block type II Mobitz I and II, 3 had LBBB, and 1 was subjected to the procedure of the persistent foramen ovale catheter occlusion. When performing the analysis according to Jacquier's method, PMs were visually identified and considered sufficiently separated from trabeculae in 41 out of the 77 examinations–LVNC in 10 (24%) patients, nDCM in 11 (73%) patients, the control group in 20 (100%) subjects; (p<0.001). The whole LVNC group differed from the control in all analyzed parameters (LV volumes, masses, and EF; p<0.001) but age, BSA, and sex (p>0.050)–see Table 4. The differences between the whole LVNC and nDCM groups only concerned left ventricular volumes: EDV, ESV, EDV$_H$, and NCM$_J$/LVM$_J$–see Table 4. In consequence, the only parameters which differentiated both the nDCM and the control group from the whole LVNC were EDV, ESV, EDV$_H$, and NCM$_J$/LVM$_J$. It should be noted that EF in the LVNC group was

**Table 4. Comparison of the whole left ventricular noncompaction group (LVNC) with the non-ischemic dilative cardiomyopathy (nDCM) and the control group.**

| Parameter | LVNC (N = 42) | nDCM (N = 15) | $P_{value\ LVNC\ vs\ DCM}$ | Control (N = 20) | $P_{value\ LVNC\ vs\ Control}$ |
|---|---|---|---|---|---|
| Age [y] | 45 (±17) | 45 (±19) | 0.957 | 49 (±19) | 0.523 |
| BSA [m2] | 1.8 (±0.17) | 1.88 (±0.26) | 0.208 | 1.92 (±0.29) | 0.059 |
| Sex (male) | 20 (47.6%) | 10 (69.2%) | 0.205** | 11 (54.4%) | 0.587** |
| EDV [ml] | 221 (173–73) | 281 (±60) | 0.038* | 117 (± 30) | <0.001* |
| $EDV_H$ [ml] | 178 (±62) | 221 (±55) | 0.028 | 96 (±27) | <0.001 |
| ESV [ml] | 146 (97–221) | 218 (±71) | 0.035* | 49 (±18) | <0.001* |
| EF [%] | 31(±12) | 24(±10) | 0.055 | 59 (±7) | <0.001 |
| $LVM_H$ [g] | 174 (148–225) | 228 (148–353) | 0.065* | 121 (±33) | <0.001* |
| $LVM_J$ [g] | 210 (175–304) | 284 (±70) | 0.494 | 145 (±38) | <0.001* |
| $NCM_H$ [g] | 53 (41–71) | 61 (35–122) | 0.347* | 22 (±5) | <0.001* |
| $NCM_J$ [g] | 119 (86–166) | 108 (61–182) | 0.656* | 34 (22–75) | <0.001* |
| $NCM_H/LVM_H$ [%] | 31 (25–34) | 27.8 (±7.3) | 0.151* | 19,0 (±4.2) | <0.001* |
| $NCM_J/LVM_J$ [%] | 41.7 (±11.1) | 27.8 (±7.2) | <0.001 | 24.1 (±10.8) | <0.001 |

Data are presented as mean ± SD (CI ±95%) and median with interquartile range (25–75%) values. BSA–body mass index (Du Bois), LV–left ventricular, EDV–LV end-diastolic volume; $EDV_H$–EDV blood corrected m. Hautvast's computed algorithm; ESV–LV end-systolic volume; $ESV_H$–ESV blood corrected m. Hautvast's computed algorithm; EF–LV ejection fraction; $NCM_J$–noncompacted layer mass m. Jacquier et al. [10]; $LVM_J$–total LV mass m. Jacquier et al. [10]; $NCM_H$–noncompacted layer mass m. Hautvast's computed algorithm [12]; $LVM_H$–total LV mass m. Hautvast's computed algorithm [12]; $NCM_J/LVM_J$–noncompacted/compacted layer mass ratio m. Jacquier et al. [10]; $NCM_H/LVM_H$–noncompacted/compacted layer mass ratio m. Hautvast's computed algorithm [12]

* p values were calculated using the U-Mann-Whitney test

** p values were calculated using the Chi-square test

markedly lower than in the control group and borderline significantly higher than in the nDCM group.

The LV trabecular distribution in the whole LVNC group is presented in Fig 3. Generally, the noncompacted to compacted layer thickness ratio gradually decreased from the apex to the base of LV. The most excessive trabeculation was observed in all apical and middle lateral and middle posterior segments of LV. In turn, none or discreetly intensified trabeculation was found in the basal and middle septal and anteroseptal segments of LV.

## The subgroup comparison

The comparison between the $LVNC_N$ subgroup and the control revealed significant differences in EDV, ESV, EF, $NCM_J$, and NCM/LVM estimated by both methods, and all of them were significantly higher in $LVNC_N$, except for the EF. There was a tendency to higher values of $EDV_H$ in $LVNC_N$.

The comparison between the $LVNC_{DCM}$ subgroup and the nDCM group revealed no significant differences in any of the analyzed parameters but $NCM_J$, $LVM_J$, and $NCM_J/LVM_J$, which were significantly higher in the $LVNC_{DCM}$. Of note, $NCM_H/LVM_H$ did not differentiate these groups. The summary of the results is presented in Tables 5 and 6.

NCM/LVM estimated by Hautvast's algorithm did not differ from Jacquier's method in the control group (19.0±4.2% vs. 24.1±11%, p = 0.164) and nDCM (27.8±7.3% vs. 27.8±7.2%, p = 0.989), but it was significantly lower in the LVNC group (30.8±7.5% vs. 41.7±11.0%, p<0.001). The difference remained significant in the LVNC subgroups: $LVNC_N$ (29.4±6.4% vs. 40.9±10.3%, p<0.001) and $LVNC_{DCM}$ (32.2±8.3% vs. 42.4±11.9%, p<0.001).

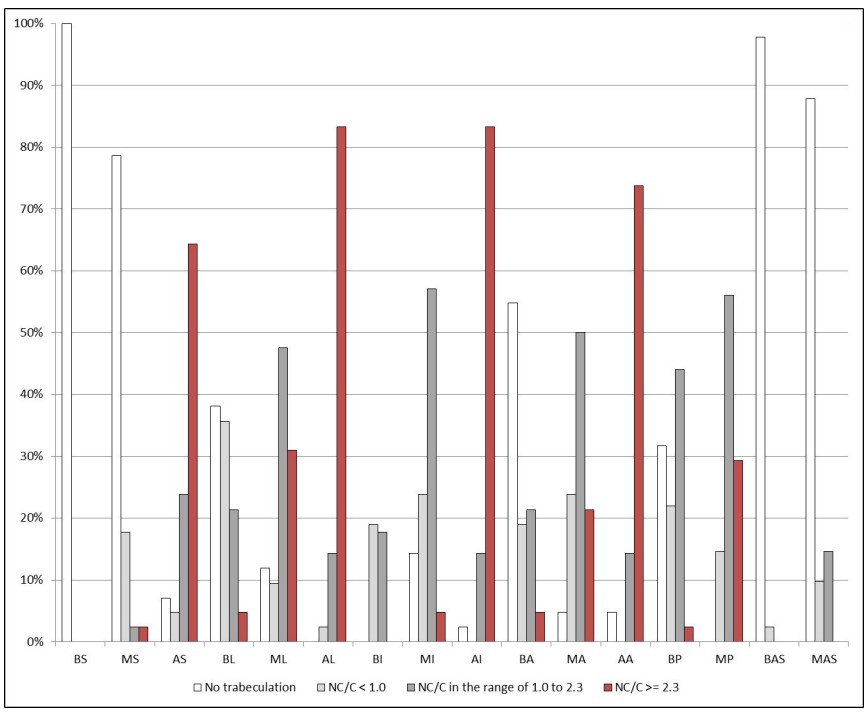

**Fig 3. The graphical distribution of the left ventricular trabeculation in the left ventricular noncompaction group.**
The vertical axis presents the percentage of the left ventricular segments with the extent of trabeculation described by
the noncompacted to compacted layer thickness ratio (NC/C) localized on the horizontal axis. The left ventricular
segments description: basal, middle, and apical septal segments (BS, MS, AS), lateral segments (BL, ML, AL), inferior
segments (BI, MI, AI), anterior segments (BA, MA, AA), basal and middle posterior segments (BP, MP) and
anteroseptal segments (BAS, MAS).

## Correlation analysis

Both the $NCM_H$ or $NCM_J$ revealed a good and similar correlation concerning EDV in the
overall examined group (r = 0.789, p<0.001 vs. r = 0.799, p<0.001; respectively), the LVNC

**Table 5. Comparison of left ventricular noncompaction subgroup with normal-range left ventricle (LVNC_N) and
the control group.**

| Parameter | LVNC_N (N = 20) | Control (N = 20) | p_post-hoc |
|---|---|---|---|
| Age [y] | 38 (±15) | 48.9 (±19.1) | 0.137 |
| BSA [m2] | 1.78 (±0.21) | 1,92 (±0.29) | 0.219 |
| EDV [ml] | 166 (±37) | 117 (±30) | 0.021 |
| EDV_H [ml] | 128 (±29) | 96 (±27) | 0.051 |
| ESV [ml] | 100 (±31) | 49 (±18) | 0.022 |
| EF [%] | 41 (±9) | 59 (±7) | <0.001 |
| LVM_H [g] | 141 (±38) | 121 (±33) | 0.376 |
| LVM_J [g] | 186 (±45) | 145 (±38) | 0.074 |
| NCM_H [g] | 41 (±15) | 22 (±5) | 0.111 |
| NCM_J [g] | 86 (37–163) | 34 (22–75) | <0.008 |
| NCM_H/LVM_H [%] | 29.4 (±6.4) | 19 (±4.2) | <0.001 |
| NCM_J/LVM_J [%] | 40.9 (±10.3) | 24 (±11) | <0.001 |

Data are presented as mean ± SD (CI ±95%) or median with interquartile range (25–75%) values. Abbreviations as in
Table 4.

**Table 6. Comparison of left ventricular noncompaction subgroup with enlarged left ventricle (LVNC$_{DCM}$) and dilated cardiomyopathy (nDCM) group.**

| Parameters | LVNC$_{DCM}$ (N = 22) | nDCM (N = 15) | P$_{post-hoc}$ |
|---|---|---|---|
| Age [y] | 52 (±16) | 45 (±19) | 0.495 |
| BSA [m2] | 1.82 (±0.13) | 1.88 (±0,26) | 0.434 |
| EDV [ml] | 300 (±71) | 281 (±60) | 0.465 |
| EDV$_H$ [ml] | 223 (±48) | 221 (±55) | 0.914 |
| ESV [ml] | 236 (±73) | 218 (±71) | 0.504 |
| EF [%] | 23 (±8) | 24 (±10) | 0.730 |
| LVM$_H$ [g] | 220 (165–447) | 228 (148–353) | 0.500 |
| LVM$_J$ [g] | 337 (±92) | 284 (±70) | 0.046 |
| NCM$_H$ [g] | 65 (41–214) | 61 (35–122) | 0.106 |
| NCM$_J$ [g] | 165 (86–317) | 108 (61–182) | 0.004 |
| NCM$_H$/LVM$_H$ [%] | 32.2 (±8.3) | 27.8 (±7.3) | 0.192 |
| NCM$_J$/LVM$_J$ [%] | 42.4 (±11.9) | 27.8 (±7.2) | 0.001 |

Data are presented as mean ± SD (CI ±95%) or median with interquartile range (25–75%) values. Abbreviations as in Table 4.

group (r = 0.800, p<0.001 vs. r = 0.816, p<0.001; respectively), and the LVNC$_{DCM}$ subgroup (r = 0.746, p<0.001 vs. r = 0.748, p<0.001; respectively); however, in the LVNC$_N$ subgroup the correlation between the NCM$_H$ and EDV was stronger in comparison to the NCM$_J$ vs. EDV (r = 0.677, p<0.001 vs. r = 0.480, p = 0.038; respectively).

In turn, in the overall examined group and the LVNC group, the NCM$_H$ or NCM$_J$ revealed a similar correlation to EF (overall examined group: r = -0.556, p<0.001 vs. r = -0.572, p<0.001, and LVNC group: r = -0.502, p<0.001 vs. r = -0.491, p<0.001; respectively). In the LVNC subgroups, no correlation between the NCM$_H$ or NCM$_J$ vs. EF was found (p>0.05).

The correlation of NCM$_H$/LVM$_H$ concerning EDV revealed a moderate correlation in the overall examined group (r = 0.434, p<0.001) and in the LVNC group (r = 0.391, p<0.01). In addition, this correlation was significant also in LVNC$_{DCM}$ subgroup (r = 0.457, p = 0.029). Correlation between NCM$_H$/LVM$_H$ and EF, though mild, was also statistically significant in the overall group (r = -0.449, p<0.001), nevertheless, the subgroup analysis revealed no correlations (LVNC: r = -0.153, p = 0.334; LVNC$_{DCM}$: r = 0.110, p = 0.618; LVNC$_N$: r = -0.143, r = 0.558).

In turn, NCM$_J$/LVM$_J$ revealed no significant correlations (see Tables A and B in S1 Table).

## Regression analysis

In the univariate regression analysis model, the NCM$_H$, NCM$_J$, and EF in similar strength explained the variability of EDV. In turn, only the NCM$_H$/LVM$_H$, of the two NCM/LVM estimation methods, explained the variability of EDV (F = 17.50, p <0.05) (see Table A in S2 Table).

In the multivariate stepwise regression analysis model (factors included: EF, and NCM$_H$, or NCM$_J$, or NCM$_H$/LVM$_H$), both the EF and NCM$_H$ or EF and NCM$_J$ models similarly explained the variability of EDV (see Table B in S2 Table).

## Cut-off values of noncompacted mass measurements between the LVNC and the control groups (ROC analysis)

The comparison of the cut-off values is presented in Table 7. Concerning the NCM/LVM, both methods with similar sensitivity and specificity differentiate the LVNC and the control

**Table 7. The cut-off values of left ventricular noncompaction mass (NCM) between the left ventricular noncompaction group (LVNC) and the control group–ROC analysis.**

| Parameter | Cut-off value | AUC | Sensitivity | Specificity |
|---|---|---|---|---|
| $NCM_H$ | 26g | 0.955 | 92.9% | 90.9% |
| $NCM_J$ | 39.9g | 0.944 | 95.2% | 72.7% |
| $NCM_H/LVM_H$ | 22% | 0.933 | 95.2% | 81.8% |
| $NCM_J/LVM_J$ | 26% | 0.883 | 95.2% | 81.8% |

$NCM_H$–noncompacted layer mass m. Hautvast's computed algorithm [12]; $NCM_J/LVM_J$–noncompacted/compacted layer mass ratio m. Jacquier et al. [10]; $NCM_H/LVM_H$–noncompacted/compacted layer mass ratio m. Hautvast's computed algorithm [12]; AUC–area under the ROC curve

group; however, the classifier $NCM_H/LVM_H$ appeared to better differentiate these two groups. In turn, the absolute $NCM_H$ seemed to have a better specificity in comparison to $NCM_J$.

### Inter- and intra-observer variability

The results of reproducibility and repeatability are presented in Table 8. Intra- and inter-observer variability of Hautvast's method was more reproducible and repeatable in comparison to Jacquier's approach.

## Discussion

The presented approach to NCM measurement utilizing Hautvast's computed algorithm method has shown excellent reproducibility and repeatability compared to Jacquier's approach [10,12]. It differentiated the LVNC group from the controls with a higher specificity considering $NCM_H$ in comparison to $NCM_J$ and might be especially applicable in the $LVNC_N$ subgroup. Comparing to the NCM measurement method based on Hautvast's computed algorithm, $NCM_J/LVM_J$ neither correlated with EDV or EF nor explained the EDV variability.

The results of our study pointed to the importance and necessity of the automation, standardization, and selection of the NCM measurement method. The estimation of NCM and NCM/LVM should be interpreted with due consideration of the methodology that was applied. In our study, the cut-off values for LVNC recognition related to $NCM_H/LVM_H$ and $NCM_H$ were lower than for Jacquier's method and also higher than the cut-off value of $NCM_J/LVM_J$ presented in Jacquier's research [10]. This is in line with doubts regarding the clinical efficacy of Jacquier's method in recognition of LVNC [5]. It also raises the question of whether the percentage of $NCM_H$ may indirectly (through EDV and/or EF) influence the prognosis in LVNC. The lack of influence of the LV trabeculae on the adverse cardiovascular outcomes in

**Table 8. Comparison of the inter- and intra-observer variability between the method based on Hautvast's computed algorithm and Jacquier's approach of noncompacted myocardial mass evaluation.**

| Parameter | Inter-observer variability (reproducibility) | | | Intra-observer variability (repeatability) | | |
|---|---|---|---|---|---|---|
| | CoV | ICC | ICC's ±95CI | CoV | ICC | ICC's ±95CI |
| $NCM_H$ | 4.3% | 0.998 | 0.990 to 0.999 | 3.7% | 0.998 | 0.991 to 0.999 |
| $NCM_H/LVM_H$ | 4.3% | 0.981 | 0.919 to 0.996 | 4.9% | 0.978 | 0.896 to 0.995 |
| $NCM_J$ | 20.5% | 0.866 | 0.552 to 0.965 | 12.8% | 0.873 | 0.268 to 0.974 |
| $NCM_J/LVM_J$ | 19.7% | 0.150 | -0.532 to 0.714 | 12,9% | 0.504 | -0.109 to 0.859 |

CoV–coefficient of variation; ICC–intraclass correlation coefficient; $NCM_J$–noncompacted layer mass m. Jacquier et al. [10]; $NCM_H$–noncompacted layer mass m. Hautvast's computed algorithm [12]; $NCM_J/LVM_J$–noncompacted/compacted layer mass ratio m. Jacquier et al. [10]; $NCM_H/LVM_H$–noncompacted/compacted layer mass ratio m. Hautvast's computed algorithm [12]

LVNC revealed in the recent studies and its clinical similarity to nDCM in terms of genetics, morphology, and clinics, often lead cardiologists and radiologists to perceive LVNC as a form of nDCM [4,16]. These doubts and questions have prompted attempts to establish the criteria of the LVNC diagnosis [17,18], which are most commonly based on Petersen's or Jacquier's observations [4,8,6]. According to Petersen et al., LVNC can be recognized if the criterion of the noncompacted to compacted layer width ratio > 2.3 is fulfilled in at least one LV segment. Thus, LVNC could be recognized in the case of the clinically silent trabeculae in the apex of a healthy LV [4,19]. In turn, the Jacquier's approach significantly overestimates the actual NCM and may lead to false conclusions regarding its influence on adverse cardiovascular events [20]. In turn, the major advantage of the presented computed method is its semi-automatic character, exclusion of the blood pool from analysis and simplification related to the unanimous inclusion of PMs into the trabecular mass. Bricq et al. previously introduced the semiautomatic assessment of the trabeculated and compacted LV mass, but the authors excluded both PMs and ITB from NCM [20,21].

It is essential to consider the precision and accuracy of the different methods of NCM analysis. Positano et al. revealed that the Grothoff's approach seemed to better capture the actual extension of trabeculated tissue than the Jacquier's, because of the exclusion of ITB volume from NCM. The inter-observer reproducibility of Grothoff's and Jacquier's methods in that study were quite similar: 9.71% and 8.22%, respectively [22]. In our study, however, the reproducibility of Jacquier's method was lower (~20%). In contrast, the simplification of the diagnostic method, related to the semi-automated blood-muscle separation, resulted in a better reproducibility of either $NCM_H$ or $NCM_H/LVM_H$ (4.3%).

In consequence, harmonizing the method of the NCM measurement, with the inclusion of PMs into NCM and the exclusion of ITB from NCM utilizing Hautvast's computed algorithm, resulted in the increased reproducibility in comparison to Jacquier's method. The increased reproducibility of the computed algorithm method was mainly achieved in terms of precision, for it is hard to improve the accuracy (trueness) of the analytic approach when there is no CMR reference method of NCM or NCM/LVM measurement. In the presented approach, the delineation of the inner and outer border of the compacted layer in short axis slices, without PMs, decreases the possible risk of error or observer-related inconsistency. The mathematical algorithm itself was operator-independent, and the risk of error was related only to movement artifacts (arrhythmia, breath-hold difficulties, etc.) and the level of cross-section slice from which the observer started or ended the LV masses analyses.

A crucial issue in clinical practice is the differentiation between LVNC with enlarged LV and nDCM. In our study, no differences in basic morphological or functional CMR parameters between $LVNC_{DCM}$ and nDCM were observed, except for $NCM_J$ and $NCM_J/LVM_J$, which should be interpreted in relation to the initial group qualification criteria. The alternative of NCM estimation used in our study, namely $NCM_H$ and $NCM_H/LVM_H$, did not differentiate these two groups. This discrepancy could be related to the differences in PMs' quantification between the methods. PMs were sufficiently separated from the trabeculae in all cases of the control and nDCM groups, in contrary to only 24% cases of the LVNC group, 13% cases of the $LVNC_{DCM}$, and 35% of the $LVNC_N$ subgroups. Thus, the mathematical algorithm used for the quantification of trabeculation may have slightly overestimated the trabeculated mass in the proportion of patients with nDCM. This, in turn, may have blurred the differentiation between nDCM and $LVNC_{DCM}$. In the case of $LVNC_N$, the presented method performed much better.

In general, LVNC recognition is mostly based on the extent of the LV hypertrabeculation. Doubts arise, however, which amount of the LV trabeculation should be considered pathological. According to Jacquier et al., LVNC could be recognized when the trabeculated LV mass was above 20% of the LV global mass [10]. Our results, however, similarly to the study of

Amzulescu et al. [4], indicate that the threshold for $NCM_J/LVM_J$ should preferably be set higher.

In contrary to Jacquier's approach, moderate correlations were found between $NCM_H/LVM_H$ and EDV in the overall examined group and the LVNC group. In turn, a significant correlation between the $NCM_H/LVM_H$ and EF was observed only in the overall group. The explanation of this may be given by Paun et al. [23]. The authors pointed at the possibly significant compensatory character of hypertrabeculation in LVNC in a malfunctioning LV, which in consequence might have an impact on stroke volume and EF [23], which in consequence might have an impact on stroke volume and EF [23] and might influence the correlation results.

## Clinical implications

Operator-independent computed algorithms of the NCM measurement, thanks to its semi-automatic character, might be a solution to increase reproducibility and repeatability, and reduce the time-consuming, operator-dependent input. The presented method might be applicable in the differentiation of LV hypertrabeculation in a non-enlarged (EDV in normal range) and at most mildly impaired (EF >40%) LV. Its possible application in case of an enlarged LV with moderate to severe dysfunction, and also the influence of the observed $NCM_H$ correlation with EDV require further research. Therefore, the estimate of NCM and NCM/LVM should be interpreted with due consideration of the methodology that was applied.

## Limitations

The qualification to the groups was based on the well-known but disputable CMR's criteria by Jacquier and Petersen, however, to increase the probability of LVNC diagnosis, we modified the cut-off value of Jacquier's method and adopted the value of 31%. As our study concentrated on the analysis of the CMR imaging, we did not relate our results to adverse clinical outcomes but the established parameters of the LV function, such as EF or EDV, as they are considered possible good prognostic factors of adverse outcomes in cardiomyopathies [1,4]. Focusing mainly on LV morphology, we did not perform any LGE or T1-mapping analysis [24].

## Conclusions

Choosing a method of NCM assessment that is less observer-dependent might increase the reliability of results. The impact of method selection on the LV parameters and cut-off values for hypertrabeculation should be further investigated.

## Supporting information

**S1 Fig. Group qualification follow chart.** The stages of group qualification are marked with *italics*. LVNC–left ventricular noncompaction; nDCM–nonischemic dilated cardiomyopathy; Control–control group; pts.–patients.
(PDF)

**S1 Table. Pearson's correlation between left ventricular noncompaction mass and left ventricular end-diastolic volume or ejection fraction.** $NCM_H$–noncompacted layer mass m. Hautvast's computed algorithm [12]; $NCM_J$–noncompacted layer mass m. Jacquier et al. [10]; $NCM_J/LVM_J$–noncompacted/compacted layer mass ratio m. Jacquier et al. [10]; $NCM_H/LVM_H$–noncompacted/compacted layer mass ratio m. Hautvast's computed algorithm [12];

EF–left ventricular ejection fraction; EDV–left ventricular end-diastolic volume.
(DOCX)

**S2 Table. Regression analysis models.** NCM$_H$–noncompacted layer mass m. Hautvast's computed algorithm [12]; NCM$_J$–noncompacted layer mass m. Jacquier et al. [10]; NCM$_J$/LVM$_J$–noncompacted/compacted layer mass ratio m. Jacquier et al. [10]; NCM$_H$/LVM$_H$–oncompacted/compacted layer mass ratio m. Hautvast's computed algorithm [12]; EF–left ventricular ejection fraction; EDV–left ventricular end-diastolic volume.
(DOCX)

## Author Contributions

**Conceptualization:** Marcin Kubik, Alicja Dąbrowska-Kugacka, Karolina Dorniak.

**Data curation:** Marcin Kubik, Marta Kutniewska-Kubik.

**Formal analysis:** Marcin Kubik, Marta Kutniewska-Kubik.

**Investigation:** Marcin Kubik, Alicja Dąbrowska-Kugacka, Karolina Dorniak.

**Methodology:** Marcin Kubik, Alicja Dąbrowska-Kugacka, Karolina Dorniak, Marta Kutniewska-Kubik.

**Project administration:** Alicja Dąbrowska-Kugacka.

**Resources:** Marcin Kubik, Alicja Dąbrowska-Kugacka, Karolina Dorniak, Ludmiła Daniłowicz-Szymanowicz, Ewa Lewicka.

**Software:** Karolina Dorniak.

**Supervision:** Alicja Dąbrowska-Kugacka.

**Validation:** Karolina Dorniak.

**Visualization:** Marcin Kubik, Marta Kutniewska-Kubik.

**Writing – original draft:** Marcin Kubik, Alicja Dąbrowska-Kugacka, Karolina Dorniak.

**Writing – review & editing:** Alicja Dąbrowska-Kugacka, Karolina Dorniak, Ludmiła Daniłowicz-Szymanowicz, Ewa Lewicka, Edyta Szurowska, Grzegorz Raczak.

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
