## [Decision Letter · Decision Letter 0]

18 Dec 2019

PONE-D-19-30605

Relationship between left ventricular hypertrabeculation mass, left ventricular volume and ejection fraction - comparison between manual and semiautomatic CMR image analysis methods

PLOS ONE

Dear Mr. Kubik,

Thank you for submitting your manuscript to PLOS ONE. After careful consideration, we feel that it has merit but does not fully meet PLOS ONE’s publication criteria as it currently stands. Therefore, we invite you to submit a revised version of the manuscript that addresses the points raised during the review process.

Authors should address all raised questions.

We would appreciate receiving your revised manuscript by Feb 01 2020 11:59PM. To enhance the reproducibility of your results, we recommend that if applicable you deposit your laboratory protocols in protocols.io, where a protocol can be assigned its own identifier (DOI) such that it can be cited independently in the future. For instructions see: http://journals.plos.org/plosone/s/submission-guidelines#loc-laboratory-protocols

A rebuttal letter that responds to each point raised by the academic editor and reviewer(s). This letter should be uploaded as separate file and labeled 'Response to Reviewers'. 

A marked-up copy of your manuscript that highlights changes made to the original version. This file should be uploaded as separate file and labeled 'Revised Manuscript with Track Changes'.

An unmarked version of your revised paper without tracked changes. This file should be uploaded as separate file and labeled 'Manuscript'.

We look forward to receiving your revised manuscript.

Kind regards,

Otavio Rizzi Coelho-Filho, M.D., Ph.D., M.P.H.

Academic Editor

PLOS ONE

Journal Requirements:

2. Please provide the full name of the ethics committee which approved this study in your methods section.

3. Please specify in the ethics statement in the Methods section and online submission information whether the participants provided informed verbal or informed written consent for inclusion in this study. If consent was verbal, please amend your current ethics statement to explain 1) why written consent was not obtained, 2) how you recorded/documented participant consent, 3) whether your ethics committee approved this consent procedure.

*For reproducibly purposes, please provide further information on the medical database records used in this study and explain how researchers may access them.

Additional Editor Comments (if provided):

Drs. Kubik and coworkers investigated the accuracy of different methods on left ventricular volumes and ejection fraction (EF) to identify noncompacted myocardial mass (NCM). Left ventricular noncompaction (LVNC) is an important condition that faces several diagnostics challenges and the current study aimed to explore novel methods to improve its identification.

Recently T1 mapping characterization with ECV quantification has been shown to provide completará information to LV volumes and LGE (Eur Heart J Cardiovasc Imaging. 2018 Aug 1;19(8):888-895. doi: 10.1093/ehjci/jey022.). I wonder if T1 mapping is available.

Reviewers' comments:

Reviewer's Responses to Questions

**Comments to the Author**

1. Is the manuscript technically sound, and do the data support the conclusions?

Reviewer #1: Partly

Reviewer #2: Partly

2. Has the statistical analysis been performed appropriately and rigorously? 

Reviewer #1: Yes

Reviewer #2: N/A

3. Have the authors made all data underlying the findings in their manuscript fully available?

Reviewer #1: Yes

Reviewer #2: Yes

4. Is the manuscript presented in an intelligible fashion and written in standard English?

Reviewer #1: Yes

Reviewer #2: Yes

5. Review Comments to the Author

Reviewer #1: The article entitled “Relationship between left ventricular hypertrabeculation mass, left ventricular volume and ejection fraction – comparison between manual and semiautomatic CMR image analysis methods” investigates a very relevant topic related to an increasingly diagnosed condition, left ventricular noncompaction cardiomyopathy(LVNCC). Some points need explanations and are listed below.

1. INTRODUCTION

a. Page 3, line 50 – You state that LVNC is ‘the” unique inherited cardiomyopathy. Many other inherited cardiomyopathy have been described, and no clear evidence of an exclusive genetic background is available. I would prefer to say “A” unique…., although its morphological features may appear in other conditions and genetic mutations either1.

b. Page 3, Line 56 and 57 _ The mentioned reference(yours ref.1) states clearly about the presence of Late Gadolinium Enhancement as a predictor, better than LVEF. So it may be modified as a suggestion.

c. Page 4, Line 80-81 _ You state here that a modification in the Phillips’ approach was made to include papillary muscles in the non-compacted myocardium (NCM). This is a methodological information, presented again in the Methods section, page 9, lines 174-175. It is important to clarify if this correction was automatic or manually performed.

2. METHODS

a. Page 8, line 134 _ CLM abbreviation is first cited here but it significance is found only in Line 192 part of Table 3. Please correct it.

b. Page 10, lines 178 and 179 _ Here you refer to the figures of the two methods used. My suggestion is to refer each figure close to the description of each method, so the readers can correlate easily.

c. Also, a detailed legend of figures 1 and 2 is presented in lines 194 to 204. It seems misplaced. A legend page is needed.

d. Page 10,Table 3 _ what is the meaning of the asterisk? It should be mentioned in the legend.

e. Page 12, line 208 _ the zero after 5 is not necessary. Please exclude.

3. RESULTS

a. Page 13, first paragraph _ in the selection process you excluded 11 of 31 controls. This seems to be related to your confidence interval criteria (18-31%), maybe being to strict. Probably a comment is necessary.

b. Page 14 _ Please clarify the meaning of the asterisk signal in Table 4.

c. Page 14 _ table 4 and lines 249-250. An important point is that the Philips method could not differentiate dilated cardiomyopathies from noncompaction cardiomyopathy, so it may not be clinical applicable. Can you comment on that?

d. Page 16, lines 274 -281 _ Please exclude these lines from the main text and include in the legends page.

e. You present in your tables 4, 5 and 6 two measurements for EDV. One obtained from traditional manual tracing and the second one from Philips software. Why are they so different? From Table 3 definitions, in the Philips algorhytm the endocardial exclusion of papilary muscles only , LVNCdcm decreased 80mL and nearly 40mL in LVNCn.

f. Page 19, lines 309-310 _ Correlation between NCMph/LVMph and EF wasat most modest (r=0.44) although significant but only for the overall group. I suggest presenting all correlations since no data from Jaquier’s method is presented. So, your conclusions at Page23-24 can not be corroborated by the evidence presented.

g. Page 19, lines 313-315 _ The cut-off values are isolated with not context. Plase expand the explanation of these values presented.

h. Page 19, Table 7 _ Can you provide an explanation for the interobserver value of 9.9% for NCM, since it is a semiautomatic method? Since the same slice was used in the comparative analysis and the explanation provided in Pages 393-395 do not apply.

1. Monserrat L, Hermida-Prieto M, Fernandez X, Rodriguez I, Dumont C, Cazon L, Cuesta MG, Gonzalez-Juanatey C, Peteiro J, Alvarez N, Penas-Lado M and Castro-Beiras A. Mutation in the alpha-cardiac actin gene associated with apical hypertrophic cardiomyopathy, left ventricular non-compaction, and septal defects. European heart journal. 2007;28:1953-61.

Reviewer #2: LVNC is a heterogeneous condition with many gaps on its etiopathogenesis, propaedeutic and treatment. One of the major concerns in LVNC is the growing of number of false-positive cases diagnosed by imaging in the recent years. The strength of this well-written paper is to demonstrate a higher reproducibility of a new computed algorithm for the quantification of degree of LV trabeculation when compared with Jaquier’s method (which indeed suffers from the reader variability and may affect the diagnosis). However, I’m concerned with the author’s aims/conclusions regarding the relationship between this new quantification and LV remodeling. As acknowledged by the authors, evidences from the largest cohorts of LVNC patients indicated the degree of LV trabeculation seems to have no prognostic impact on cardiovascular events (neither Petersen’s nor Jaquier’s method). Other studies investigating extracellular matrix by CMR T1 mapping (Araujo-Filho et al, EHJ, 2018) and microcirculation/metabolism by PET (Jenni et al, JACC, 2002; Tavares-Melo et al, EHJ, 2017) failed to find a significant relationship with the amount of trabeculations. These support that the degree of LV trabeculation is not a mediator of adversity. The authors indeed show a statically significant (but clinically questionable) correlation between EDV and LV trabeculation using the algorithm. This is expected once the source of variability of the Jaquiers’s method is removed, but the authors should be careful with the potential prognostic impact of this new approach and may highlight more its potential diagnosis role.

There are other issues that need to be addressed:

# “23 examinations were excluded from further analysis due to death before the study qualification, doubtful diagnosis, CMR artifacts and additional cardiac diseases which could influence the group qualification”. Why did the control group have the largest proportional exclusion (from 31 to 21)? Would this affect the found cut-off value?

# “...in the LVNC group 2 individuals had prior myocardial inflammation”. The authors mean previous myocarditis before diagnosis of LNVC? If yes, how confident are they with the diagnosis of LVNC in these 2 cases?

# “Independent t-Student and U-Mann-Whitney tests were used where appropriate”. The authors should review the statistical approach for multiple comparison in the subgroup analysis.

# “The Chi-squared test was used for nonparametric data”. Please review this statement since chi-squared test is used to determine whether there is a significant difference between the expected frequencies and the observed frequencies in one or more categories.

# “In order to determine the cut-off value for the pathological trabecular mass in our population

according to Jacquier's method, apart from the mean ± SDs of NCMJ/LVMJ, the upper confidence interval (+95% CI) was assessed in our control group, following Choi et al”. Those cited authors seems to have used receiver operating characteristics (ROC) curves to determine the optimal cut-off values. Please review the reference.

# “Finally, a multivariate regression analysis model was created to estimate the potential influence of the examined parameters on EF and EDV”. Please provide the independent variables which were included in the model.

# “The F-Snedecor test and t-Student test were used to compare the accuracy of the two analyzed methods.”. Please review if these tests are appropriate to investigate accuracy.

6. PLOS authors have the option to publish the peer review history of their article (what does this mean?). If published, this will include your full peer review and any attached files.

Reviewer #1: No

Reviewer #2: No

---

## [Author Response · Author response to Decision Letter 0]

1 Feb 2020

We would like to express our thanks for a thorough and detailed revision of our manuscript. Please, find answers to Reviewers below.

I. Responses to the Editor:

1) Journal Requirements

We did our best to ensure that our manuscript meets PLOS ONE’s requirements. We re-verified the compliance of the paper with PLOS ONE’s standard and corrected incompatibilities.

2) Ethics Committee Board Full Name

We provided information on the full institutional research ethics board name and the number of consent in the section “Materials and Methods”.

3) Specifying in the ethics statement the type of consent provided by the study participants

In the “Materials and Methods” section, we specified the ethics statement by adding information on the type of consent (informed written consent) provided by the study participants.

4) For reproducible purposes, please provide further information on the medical database records used in this study and explain how researchers may access them

At the beginning of the section “Materials and Methods”, we added information on how to obtain data from the medical database in our medical center. Generally, to obtain access to the database researchers have to receive written consent of heads of departments of radiology and of cardiology. Subsequently, final approval is given by the principal director of the University Clinical Centre of the Medical University of Gdańsk in Poland. Fully deidentified data are available upon request from the corresponding author.

5) Recently T1 mapping characterization with ECV quantification has been shown to provide complete information to LV volumes and LGE. I wonder if T1 mapping is available?

We consider the CMR T1-mapping technique an inspiring option in diagnostics of left ventricular noncompaction, as well as the well-documented late gadolinium enhancement imaging (LGE). However, our study verified a potential superiority of automated radiological evaluation of the extent of left ventricular (LV) noncompacted mass (NCM) and its possible influence on LV end-diastolic volume (EDV) and LV ejection fraction (EF). LGE might improve risk stratification when added to clinical and morphological criteria for LVNC. Nevertheless, in our study, LGE and T1-mapping techniques were not considered.

 

I. Responses to Reviewer 1:

1) INTRODUCTION

COMMENT_1: Page 3, Line 50 – You state that LVNC is ‘the” unique inherited cardiomyopathy. Many other inherited cardiomyopathy has been described, and no clear evidence of an exclusive genetic background is available. I would prefer to say “A” unique…., although its morphological features may appear in other conditions and genetic mutations either.

RESPONSE_1: We agreed with the opinion of Reviewer 1 on the difference between: „a unique inherited cardiomyopathy” and „the unique…”. As it was mentioned, this particular cardiomyopathy does not have a corresponding specific genotype distinguishing it from the others. According to the above, we changed the words “the unique…” to “a unique…” in the introduction section of our manuscript.

COMMENT_2: Page 3, Lines 56 and 57 – The mentioned reference (your ref. 1) states clearly about the presence of Late Gadolinium Enhancement as a predictor, better than LVEF. So it may be modified as a suggestion.

RESPONSE_2: We fully agree that the role of LGE in different cardiac diseases (i.e., myocarditis, cardiomyopathies, coronary heart disease) is undoubtful and well-described in the medical literature. Although it was recently associated also with LVNC, it seems to be a feature of all cardiomyopathies.

The authors of the ref. 1 clearly stated that “detection of LV fibrosis is a robust independent predictor of poor prognosis.” However, they also clearly pointed to a high rate of cardiovascular events (CEs) in patients with LV dilation and reduced EF - a phenotype of LVNC classified as dilated cardiomyopathy (DCM)-like, the same we analyzed in our study. With regard to EF and EDV as predictors of CEs, the authors referred to the research by Mavrogeni et al. (2012). At univariate analysis, they found EF, EDV, and LGE as independent predictors of CEs. Subsequently, in a multivariate analysis, LGE was the only independent predictor of CEs. Nevertheless, none of the three presented models consisted of both EF and EDV; moreover, no head-to-head improvement analysis was made between EF vs. EDV vs. LGE. In all presented cases, LGE was only additional to EF or EDV; however, it significantly improved the risk assessment.

According to the above mentioned, we have modified our statement. Moreover, we added a reference (no 4; Amzulescu MS et al., JACC Cardiovasc Imaging. 2015;8(8):934-46. doi: 10.1016/j.jcmg.2015.04.015). In this reference, the authors stated that the prognosis of patients with LVNC is mainly affected by the presence of heart failure symptoms (clinical condition), LV dilatation, and systolic dysfunction; however, the presence and extent of LGE were also statistically significant predictors of outcome in their examined population.

COMMENT_3: Page 4, lines 80 and 81 – You state here that a modification in the Phillips’ approach was made to include papillary muscles in the non-compacted myocardium (NCM). This is methodological information, presented again in the Methods section, page 9, lines 174-175. It is important to clarify if this correction was automatic or manually performed.

RESPONSE_3: We agree that the information about the inclusion of papillary muscles in the noncompacted myocardium is a methodological issue, and it was repeated in the section methodology line 165-166; therefore, we have modified the manuscript accordingly. In the methodology section, an explanation was added that after manual correction of the endocardial trace, the process of inclusion of papillary muscles into noncompacted mass was automatic.

2) METHODS and RESULTS

COMMENT_4: Page 8, Line 134 – CLM abbreviation is first cited here but its significance is found only in Line 192 part of Table 3. Please correct it

RESPONSE_4: With all due respect to Reviewer, the first description of the CLM abbreviation was placed on page 3, line 71 of the manuscript.

COMMENT_5: Page 10, Lines 178 and 179 – Here, you refer to the figures of the two methods used. My suggestion is to refer each figure close to the description of each method so that the readers can correlate easily.

RESPONSE_5: According to the Reviewer’s suggestion, the references to figures 1 and 2 were placed near the description of each method.

COMMENT_6: Also, a detailed legend of figures 1 and 2 is presented in lines 194 to 204. It seems misplaced. A legend page is needed.

RESPONSE_6: We have corrected the location of tables and figures’ legends according to submission guidelines.

COMMENT_7: Page 10, Table 3 – What is the meaning of the asterisk? It should be mentioned in the legend.

RESPONSE_7: The asterisk in Table 3 marked an observer-dependence. It is now explained in the legend of Table 3.

COMMENT_8: Page 12, Line 208 – The zero after 5 is not necessary. Please exclude.

RESPONSE_8: Corrected.

3) RESULTS

COMMENT_9: a) Page 13, 1st paragraph – In the selection process, you excluded 11 of 31 controls. This seems to be related to your confidence interval criteria (18-31%), maybe being too strict. Probably a comment is necessary.

RESPONSE_9: Indeed, only 20 of the 31 controls were included in the analysis process. This was due to the fact that as we retrospectively sought for the controls, we primarily enrolled individuals who were initially evaluated as “no cardiac disease.” (31 controls). However, upon careful reviewing their medical history and CMR examinations, we excluded individuals whose morphological features and/or further clinical follow-up were found significant (e.g., borderline myocardial hypertrophy, coronary artery disease found in the course of further testing or medical history suggesting other systemic diseases, etc.) so that the control group was unequivocally healthy. Hence 20 controls were finally considered.

As a result, five controls were excluded from further analysis due to morphological considerations upon CMR review, three controls due to cardiovascular disease confirmed upon further evaluation (i.e., significant coronary artery disease), two controls due to significant imaging artifacts (precluding adequate LVM estimation), and one control due to death prior to the beginning of the study. Hence, we believe that this qualification criterion, though it might seem strict, is probably adequate, based on the careful review of the control group.

Accordingly, we added the information clause: “The study was planned and performed following the European Association of Cardiovascular Imaging (EACVI) cardiac diagnostics guidelines and the Polish National Health Fund. Thus, the CMR scans were performed as part of the standard out- and inpatients cardiac diagnostic process.” to the first paragraph of the “Materials and Methods” section and supplement the manuscript with the figure facilitating understanding the group enrollment process.

COMMENT_10: Page 14 – Please clarify the meaning of the asterisk signal in Table 4.

RESPONSE_10: The asterisk in Table 4 marked the level of significance of comparisons by the U-Mann-Whitney test, and (**) by the Chi-square test. The meaning of the asterisk, as well as double asterisk, was explained in the legend of Table 4.

COMMENT_11: Page 14, Table 4 and lines 249 and 250 – The important point is that the Philips method could not differentiate dilated cardiomyopathies from noncompaction cardiomyopathy, so it may not be clinically applicable. Can you comment on that?

RESPONSE_11: To our knowledge, it is not easy to differentiate between LVNC and nDCM due to (i) overlapping syndrome LVNC/nDCM (DOI: 10.12659/MSM.909172), genetic and clinical similarities (DOI: 10.17219/acem/67457), and possible changing in phenotype (DOI: 10.1016/j.yjmcc.2010.03.00). In turn, one of the assumptions of our research was to verify if the automatic method of measurement of the extent of NCM may potentially influence the LVNC diagnosis in comparison to the one presented by Jacquier et al., in the context of the new approach to NCM evaluation proposed recently by Contour et al. We used the software utilizing the method of LV masses assessment by Hautvast et al. available at our center (not introduced to LVNC diagnosis before).

Our study clearly shows that our method easily differentiates between the LVNC group and the controls. Doubts arise with regard to differentiation between the LVNC with enlarged LV subgroup (LVNCDCM) and nDCM. In our opinion, this problem requires a critical approach and discussion because of the frequent use of Jacquier’s method in researches on LVNC.

The method we propose is less dependent on the observer, and thus, the risk of NCM measurement error is reduced. In turn, it might help differentiate LVNC from other clinical conditions and standardize measurements between clinical centers.

However, the differentiation between LVNCDCM and nDCM is problematic, and the clinical similarity of the two cardiomyopathies, in the light of continued doubts about the impact of LV hypertrabeculation itself on disease outcome, raises a diagnostic and therapeutic question of whether LVNCDCM and nDCM are to be differentiated at all.

As we were probably the first to use Hautvast’s algorithm to differentiate between LVNC and the controls or nDCM, the potential clinical application requires further research. It is also possible that, in combination with the clinical symptoms, and LGE or T1 mapping, Hautvast’s algorithm might potentially contribute to better cardiovascular risk stratification in LVNC. 

COMMENT_12: Page 16, Lines 274 - 281. Please exclude these lines from the main text and include them in the legends page.

RESPONSE_12: In accordance with the PLOS ONE’s guidelines, we have placed all tables immediately after the paragraph of their first citation. The tables’ legends are located right under the appropriate tables.

COMMENT_13: You present in your Tables 4, 5 and 6 two measurements for EDV. One obtained from traditional manual tracing and the second one from Philips software. Why are they so different? From Table 3 definitions, in the Philips algorithm, the endocardial exclusion of papillary muscles only, LVNCDCM decreased 80mL and nearly 40mL in LVNCN.

RESPONSE_13: The difference between the two methods is related not only to LV papillary muscles. The presented Hautvast’s algorithm counts voxels according to specific signal intensity. In normal conditions, the border between layers is clearly visible thanks to not excessive trabeculation and well-developed (not fragmented) papillary muscles. The difference between the two end-diastolic volumes is about 20-25ml, what is mostly dependent on the papillary muscles’ volume. The difference, however, increases with papillary muscles fragmentation and the extend of trabeculation (note that the LVNC group had the most excessively fragmented papillary muscles, most excessive trabeculation, and technically speaking, artifacts were also visible mostly in the LVNC group, though it was hard to compare the examined groups in terms of artifacts). The volume difference is probably dependent on (i) intracavitary blood artifacts, (ii) blurred border between muscle tissue and blood due to artifacts, (iii) smaller voxel intensity difference between the tissue of the trabeculae or fragmented papillary muscle parts due to very intensive trabecular net and narrow intertrabecular recesses, and (iv) blurred interlayer border due to excessive and irregularly penetrating LV compacted layer tissue recesses. Especially in situations mentioned above the error related to possible improper voxel accounts in favor of muscle tissue may significantly decrease EDV estimated by the Hautvast’s algorithm (algorithm dependence). The volume difference was also greater In the presence of trabecular net morphology of the LV only seen in the LVNC group, especially in LVNCDCM. We think that our hypotheses on the significance of these observations on results should be taken into account in further studies.

COMMENT_14: Page 19, Lines 309-310 – Correlation between NCMph/LVMph and EF was at most modest (r=0.44) although significant but only for the overall group. I suggest presenting all correlations since no data from Jaquier’s method is presented. So, your conclusions on Page 23-24 can not be corroborated by the evidence presented.

RESPONSE_14: According to Reviewer’s suggestion, we added information on correlations other than NCMPh/LVMPh and EF.

COMMENT_15: Page 19, lines 313-315. The cut-off values are isolated with no context. Please expand the explanation of these values presented.

RESPONSE_15: The cut-off values for the extent of NCM in LVNC serve to compare them with the respective cut-off values of Jacquier’s method. Therefore, we supplemented the paragraph with the respective cut-off values, according to Jacquier's approach, related to the examined population, as well as to the values presented in Jaquier’s original publication.

COMMENT_16: Page 19, Table 7. Can you provide an explanation for the interobserver value of 9.9% for NCM, since it is a semiautomatic method? Since the same slice was used in the comparative analysis and the explanation provided in Pages 393-395 do not apply.

RESPONSE_16: We would like to point out that the whole assessment of the NCM and NCM/LVM was based on a stack of short-axis slices so the risk of error was additively increased, and the NCM value of 9.9% related to Jacquier method which is more observer-dependent. In the revised version of the manuscript, according to the comment of Reviewer 2, we changed the method of intra- and interobserver variability measurement. The repeatability of NCMJ measurement, however, remained very low. Additionally, to improve the readability of the manuscript and distinction of the two methods quoted therein, we decided to unify the designations by changing the name of the Philips’s to Hautvast’s method, noting that Hautvast et al. first described the algorithm, while it is only implemented into the Philips software. 

 

 II. Responses to Reviewer 2:

COMMENT_1: I’m concerned with the author’s aims/conclusions regarding the relationship between this new quantification and LV remodeling. As acknowledged by the authors, evidence from the largest cohorts of LVNC patients indicated the degree of LV trabeculation seems to have no prognostic impact on cardiovascular events (neither Petersen’s nor Jaquier’s method). Other studies investigating extracellular matrix by CMR T1 mapping (Araujo-Filho et al, EHJ, 2018) and microcirculation/metabolism by PET (Jenni et al, JACC, 2002; Tavares-Melo et al, EHJ, 2017) failed to find a significant relationship with the amount of trabeculations. This supports that the degree of LV trabeculation is not a mediator of adversity. The authors indeed show a statistically significant (but clinically questionable) correlation between EDV and LV trabeculation using the algorithm. This is expected once the source of variability of the Jaquiers’s method is removed, but the authors should be careful with the potential prognostic impact of this new approach and may highlight more its potential diagnosis role.

RESPONSE_1: The aim of our study was to compare Jacquier’s method (as one of the currently used methods of NCM measurements) to the computed algorithm by Hautvast et al. We demonstrated that the degree of observer-dependency may affect the results obtained. We have considered EDV and EF as important parameters related to adverse outcomes in many cardiac diseases, also cardiomyopathies. In conclusion, we demonstrated the possible consequences of choosing one approach vs. the other. To highlight a more diagnostic role of our paper, we drew attention to this fact in the conclusions section.

COMMENT_2: “23 examinations were excluded from further analysis due to death before the study qualification, doubtful diagnosis, CMR artifacts and additional cardiac diseases which could influence the group qualification”. Why did the control group have the largest proportional exclusion (from 31 to 21)? Would this affect the found cut-off value?

RESPONSE_2: Indeed, only 20 of the 31 controls were included in the analysis process. This was due to the fact that as we retrospectively sought for the controls, we primarily enrolled individuals who were previously evaluated as “no cardiac disease.” (31 controls). However, upon careful reviewing their medical history and CMR examinations, we excluded individuals whose morphological features and/or further clinical follow-up were found significant (e.g., borderline myocardial hypertrophy, coronary artery disease found in the course of further testing or medical history suggesting other systemic diseases, etc.) so that the control group was unequivocally healthy. Hence 20 controls were finally considered.

As a result, five controls were excluded from further analysis due to morphological considerations upon CMR review, three controls due to cardiovascular disease confirmed upon further evaluation (i.e., significant coronary artery disease), two controls due to significant imaging artifacts (precluding adequate LVM estimation), and one control due to death prior to the beginning of the study. Hence, we believe that this qualification criterion, though it might seem strict, is probably adequate, based on the careful review of the control group.

Accordingly, we added the information clause: “The study was planned and performed following the European Association of Cardiovascular Imaging (EACVI) cardiac diagnostics guidelines and the Polish National Health Fund. Thus, the CMR scans were performed as part of the standard out- and inpatients cardiac diagnostic process.” to the first paragraph of the “Materials and Methods” section and supplement the manuscript with the figure facilitating understanding the group enrollment process.

We want to pay your attention to the fact that not before, but after forming the control group, the NCM/LVM cut-off value of 31% was calculated. Moreover, establishing the LVNC study criterion and the manner of its calculation, we took into consideration the facts that (i) the currently used criterion of LVNC proposed by Jacquier et al. did not fully meet its task, (ii) the so-called “gold standard” criterion of the LVNC recognition has not been established yet, and we did not take into account the LGE in qualifying patients for the LVNC group as an additional criterion. Moreover, the value we adopted influenced only two individuals from the entire LVNC group, which were still clinically questionable.

COMMENT_3: “...in the LVNC group 2 individuals had prior myocardial inflammation”. Do the authors mean previous myocarditis before a diagnosis of LNVC? If yes, how confident are they with the diagnosis of LVNC in these 2 cases?

RESPONSE_3: Generally, we qualified individuals for the LVNC group by carefully revising their prior diagnosis, available medical history, and available additional test results, including echocardiography and CMR. Moreover, in the group qualification process, three independent observers took part, and medical consultations were held to ensure proper group qualification when doubts arose. The co-occurrence of LVNC and myocarditis (especially due to viral infections) is reported in the medical literature. Some case reports pointed out the possible co-existence of these two clinical conditions 1,2,3. On the other hand, some reports suggested an overlooked diagnosis of LV hypertrabeculation/LVNC or misinterpreted LV hypertrabeculation/LVNC as myocarditis 4. We also took into account the possibility of misdiagnosis of myocarditis in CMR 5. Taking the above-mentioned into account in combination with the experience of the cardiologist reading CMR images, based on abnormalities seen in a heart affected by myocarditis 6, we were convinced that the diagnosis of LVNC has been overlooked in these two cases. We would like to point out that two more cases were previously excluded from further analysis from the nDCM group due to prior myocarditis. None of them had excessive trabeculation.

1Patil KG et al. (2014). Left ventricular non-compaction with viral myocarditis: A rare presentation of a rarer disease. J Assoc Physicians India. 62:261-3.

2Oguz K et al. (2015) Which one is Worse? Acute Myocarditis and Co-existing Non-compaction Cardiomyopathy in the Same Patient. J Clin Diagn Res. 9(6): OJ01. DOI: 10.7860/JCDR/2015/11774.6033.

3Dobranici M et al. (2012). Genetic disorder or toxoplasma myocarditis: a case report of dilated cardiomyopathy with hypertrabeculation in a young asymptomatic woman. 5(1):110‐113.

4Stöllberger C et al. (2006). Pitfalls in the diagnosis of left ventricular hypertrabeculation/non-compaction. Postgrad Med J. 82:679–683. DOI: 10.1136/pgmj.2006.046169.

5Emrich T et al. (2015). Cardiac MR enables diagnosis in 90% of patients with acute chest pain, elevated biomarkers and unobstructed coronary arteries. Br J Radiol 88:20150025. DOI: 10.1259/bjr.20150025.

6Matthias G et al. (2013). Cardiac Magnetic Resonance Assessment of Myocarditis. Circulation: Cardiovascular Imaging. 6:833–839. DOI: 10.1161/CIRCIMAGING.113.000416.

COMMENT_4: “Independent t-Student and U-Mann-Whitney tests were used where appropriate.” The authors should review the statistical approach for multiple comparisons in the subgroup analysis.

RESPONSE_4: Being very thankful for statistical suggestions, we re-discussed the choice of the statistical tests with our statisticians and worked out that it would probably be better to perform the ANOVA testing for multiple comparisons in the subgroup analysis. Following Reviewer’s suggestion, we re-analyzed subgroups and substituted current data in Tables 5 and 6 with data obtained from the ANOVA approach and, if applicable, we have adjusted the “Results” section.

COMMENT_5: “The Chi-squared test was used for nonparametric data.” Please review this statement since the chi-squared test is used to determine whether there is a significant difference between the expected frequencies and the observed frequencies in one or more categories.

RESPONSE_5: Agreeing with Reviewer’s suggestion, we have replaced the phrase “The Chi-squared test was used for nonparametric data” with: “Differences between categorical variables were tested with the chi-square test.”

COMMENT_6: e) “In order to determine the cut-off value for the pathological trabecular mass in our population. According to Jacquier's method, apart from the mean ± SDs of NCMJ/LVMJ, the upper confidence interval (+95% CI) was assessed in our control group, following Choi et al.”. Those cited authors seem to have used receiver operating characteristics (ROC) curves to determine the optimal cut-off values. Please review the reference.

RESPONSE_6: We used Jacquier’s approach as a comparative method to Hautvast’s algorithm, but with different cut-off values obtained by the estimation of the upper confidence interval of NCM in the control group. In this context, the reference was indeed incorrect. The error was corrected in the manuscript to “Amzulescu et al.” (ref. 4).

COMMENT_7: “Finally, a multivariate regression analysis model was created to estimate the potential influence of the examined parameters on EF and EDV”. Please provide the independent variables which were included in the model.

RESPONSE_7: We have changed the paragraph and added more information on the variables we used to create the model.

COMMENT_8: g) “The F-Snedecor test and t-Student test were used to compare the accuracy of the two analyzed methods.”. Please review if these tests are appropriate to investigate accuracy.

RESPONSE_8: We are again very thankful for statistical suggestions, we re-discussed the choice of the statistical tests with our statisticians, and we worked out that it would probably be better to perform an interclass correlation test (ICC) in assessing the accuracy and to calculate the coefficient of variation. For this purpose, we repeated the statistical analysis and substituted Table 7 with new results in the Inter- and Intra-observer section.

Grateful for comments and suggestions,

The authors

---

## [Decision Letter · Decision Letter 1]

24 Feb 2020

Influence of observer-dependency on left ventricular hypertrabeculation mass measurement and its relationship with left ventricular volume and ejection fraction - comparison between manual and semiautomatic CMR image analysis methods

PONE-D-19-30605R1

Dear Dr. Kubik,

We are pleased to inform you that your manuscript has been judged scientifically suitable for publication and will be formally accepted for publication once it complies with all outstanding technical requirements.

With kind regards,

Otavio Rizzi Coelho-Filho, M.D., Ph.D., M.P.H.

Academic Editor

PLOS ONE

Additional Editor Comments (optional):

Reviewers' comments:

Reviewer's Responses to Questions

**Comments to the Author**

1. If the authors have adequately addressed your comments raised in a previous round of review and you feel that this manuscript is now acceptable for publication, you may indicate that here to bypass the “Comments to the Author” section, enter your conflict of interest statement in the “Confidential to Editor” section, and submit your "Accept" recommendation.

Reviewer #1: All comments have been addressed

Reviewer #2: All comments have been addressed

2. Is the manuscript technically sound, and do the data support the conclusions?

Reviewer #1: Yes

Reviewer #2: Yes

3. Has the statistical analysis been performed appropriately and rigorously? 

Reviewer #1: Yes

Reviewer #2: Yes

4. Have the authors made all data underlying the findings in their manuscript fully available?

Reviewer #1: Yes

Reviewer #2: Yes

5. Is the manuscript presented in an intelligible fashion and written in standard English?

Reviewer #1: Yes

Reviewer #2: Yes

6. Review Comments to the Author

Reviewer #1: (No Response)

Reviewer #2: The authors of the paper entitled "Influence of observer-dependency on left ventricular hypertrabeculation mass measurement and its relationship with left ventricular volume and ejection fraction - comparison between manual and semiautomatic CMR image analysis methods" have addressed all my comments/suggestions.

7. PLOS authors have the option to publish the peer review history of their article (what does this mean?). If published, this will include your full peer review and any attached files.

Reviewer #1: No

Reviewer #2: No

---

## [Editor Report · Acceptance letter]

27 Feb 2020

PONE-D-19-30605R1 

Influence of observer-dependency on left ventricular hypertrabeculation mass measurement and its relationship with left ventricular volume and ejection fraction –  comparison between manual and semiautomatic CMR image analysis methods 

Dear Dr. Kubik:

I am pleased to inform you that your manuscript has been deemed suitable for publication in PLOS ONE. Congratulations! Your manuscript is now with our production department. 

With kind regards,

on behalf of

Dr. Otavio Rizzi Coelho-Filho 

Academic Editor

PLOS ONE